# Examining the Effect of Freezing Temperatures on the Survival Rate of Micro-Encapsulated Probiotic *Lactobacillus acidophilus* LA5 Using the Flash Freeze-Drying (FFD) Strategy

**DOI:** 10.3390/microorganisms12030506

**Published:** 2024-03-01

**Authors:** Elsa Acosta-Piantini, Maria Carmen Villarán, Ángel Martínez, José Ignacio Lombraña

**Affiliations:** 1Department of Chemical Engineering, Faculty of Science and Technology, UPV/EHU, P.O. Box 644, 48080 Bilbao, Spain; ji.lombrana@ehu.eus; 2School of Chemical Engineering, University Autonomous of Santo Domingo (UASD), Santo Domingo 10105, Dominican Republic; angelmartinez1206@gmail.com; 3TECNALIA, Basque Research and Technology Alliance (BRTA), 01510 Vitoria-Gasteiz, Spain; mcarmen.villaran@tecnalia.com

**Keywords:** flash freeze-drying, freeze-drying, microencapsulation, *Lactobacillus acidophilus* LA5, probiotics, cell viability, rate survival, water activity, mathematical modeling

## Abstract

This work proposes a novel drying method suitable for probiotic bacteria, called flash freeze-drying (FFD), which consists of a cyclic variation in pressure (up-down) in a very short time and is applied during primary drying. The effects of three FFD temperatures (−25 °C, −15 °C, and −3 °C) on the bacterial survival and water activity of *Lactobacillus acidophilus* LA5 (LA), previously microencapsulated with calcium alginate and chitosan, were evaluated. The total process time was 900 min, which is 68.75% less than the usual freeze-drying (FD) time of 2880 min. After FFD, LA treated at −25 °C reached a cell viability of 89.94%, which is 2.74% higher than that obtained by FD, as well as a water activity of 0.0522, which is 55% significantly lower than that observed using FD. Likewise, this freezing temperature showed 64.72% cell viability at the end of storage (28 days/20 °C/34% relative humidity). With the experimental data, a useful mathematical model was developed to obtain the optimal FFD operating parameters to achieve the target water content in the final drying.

## 1. Introduction

Freeze-drying (FD) combined with microencapsulation is currently one of the most widely used methods to preserve the viability of probiotics, defined as “live microorganisms that when administered in adequate amounts confer a health benefit to the host” [1]. However, achieving cell viability high enough to maintain during commercialization remains a challenge [2]. The loss of viability of probiotics during processing, storage, and after digestion has been widely documented [3], and it has been established that any food claiming to have a probiotic effect should contain at least 10^6^ to 10^7^ CFU/mL of viable probiotic bacteria [4], as health benefits are directly related to the number of viable cells present at the time of consumption [5]. Considering the human health implications of this issue [6], as well as the economic ones, since the global probiotic market is projected to increase to around USD 110 billion by 2030, new studies should be conducted to search suitable technologies to guarantee its cell viability [7].

The mechanism used by microencapsulation to protect viability is entrapping the cells inside a matrix (alginate, chitosan, among others) to form a microenvironment where the live cells can remain alive by being protected from the damaging factors in the surroundings [8]. In addition, this process presents co-benefits as it increases the shelf life, protects against factors such as light, controls the release of the material, and improves the stability of the probiotic under in vitro gastrointestinal conditions [9] as well as during storage [10]. Innovative co-encapsulation techniques, combining different bacterial strains as well as various cryoprotective materials, have also been studied in the search for a viable long-term storage method for probiotics [11].

The strategy used by FD consists of freezing the bioactive compounds so that the water they contain is converted into ice, under a vacuum, to sublimate the ice directly into water vapor, and then extracting the water vapor. Typically, the cells are first frozen using liquid nitrogen (−196 °C) and then dried by sublimation by applying a high vacuum force [12]. In this process, three phase changes occur during the freezing and sublimation/desorption stages, so the lactobacilli must withstand many detrimental stresses (thermal, oxidation, osmotic, pH, and sub-cooling stresses, intra and extra-cellular crystallization, etc.) [13].

The freezing schematic of the freeze-drying process has an important influence on the survival of bacteria [14]; that is why one of the most critical steps in this process, to preserve cell integrity, is the freezing step, since it has been shown that cell inactivation occurs mainly in this stage. In fact, it has been shown that 60–70% of cells that survive the freezing phase can survive the dehydration process, mainly because the intracellular and extracellular solution concentrations will increase as the temperature decreases until the eutectic point is reached [15].

Therefore, one of the main factors to study to preserve the integrity of probiotic cells during drying should be the freezing rate, because if it is not high enough, it can have detrimental effects on the cells due to crystallization, which can rupture the cell membrane [16]. During slow freezing, the process of gradually dehydrating the cell as ice slowly forms outside the cell leads to extensive cell damage, while rapid freezing can avoid the effects of solute and excessive cell shrinkage [17]. Considering that freezing is a critical stage to maintain the quality of freeze-dried products, mathematical models based on energy balance have been proposed to predict the freezing rate to improve the process [18,19,20].

In this context, this work proposes a new drying method, called flash freeze-drying (FFD), which consists of a cyclic variation in pressure (up-down) in a very short time. This activation leads to the heating of the product (pressure up) and vaporization of the product’s moisture (pressure down). The objective is to evaluate the effects of three FFD temperature strategies (−25 °C, −15 °C, and −3 °C) on the bacterial survival and water activity of *Lactobacillus acidophilus* LA5, previously microencapsulated with calcium alginate and chitosan. In addition, the stabilization of the probiotic bacteria is studied during storage in darkness for 28 days at 20 °C with 34% relative humidity. Furthermore, a mathematical model is developed to obtain the optimal operating parameters for the specific water activity in the final dried product.

## 2. Materials and Methods

### 2.1. Biomass Production

*Lactobacillus acidophilus* LA5 (LA) was supplied by the Christian Hansen Laboratory of Spain as a freeze-dried culture and was stored at −80 °C until its use. The bacteria were inoculated in MRS broth (Merck, Madrid, Spain) and incubated at 37 °C for 48 h under aerobic conditions with moderate agitation (200 rpm). Then, the biomass was recovered by centrifugation at 4000 rpm at 15 °C for 15 min. A second centrifugation (4000 rpm, 15 °C, 15 min) was necessary to recover all the biomass produced. The cells obtained were washed with sterile water and kept refrigerated (5 °C) until they were used in the encapsulation process. pH analysis and total mesophilic aerobic bacteria counts in Petri dishes were also performed during biomass production. Biomass production techniques for probiotic *Lactobacillus* carried out in previous studies were applied in this work [21]. Likewise, preliminary studies were conducted to determine the optimum conditions and growth kinetics of the probiotic material.

### 2.2. Microencapsulation Process

Microencapsulation was carried out in one device for this process a jet cutter (GeniaLab, Braunschweig, Germany). Previously, to obtain the biomass of LA5, a solution at 2% (*w*/*v*) of sodium alginate (FMC Biopolymer, Girvan, UK) was prepared. Also, 1 L of a 0.2% (*w*/*v*) solution of chitosan (85% deacetylated, Sigma, Madrid, Spain) was prepared. This solution also contained 14.7 g of calcium chloride (Merck, Madrid, Spain) and 2 mL of glacial acetic acid (Merck, Spain). The pellet obtained as a description in point 2.1 was added to the alginate solution and transferred to the microencapsulation device. For the microencapsulation process, capsules were received in the chitosan solution. Equipment was operated at the following conditions: flow rate of 0.256 g/s, pressure of 1.5 bar, disk of 60 × 100, motor speed of 4000 rpm, 170-micron nozzle, and 60 × 100 disc. The capsulates were suspended in sterile milk as a cryoprotectant agent for 30 min and then drained.

To estimate the size deviation of the microcapsules, a granulometric study was performed using the optical microscope (Axioskop 40, Carl Zeiss, Oberkochen, Germany) and Ellix 5.0 software (Microvision Instruments, France).

### 2.3. Cell Viability

The cell viability of the entrapped bacteria in the alginate chitosan matrix was measure as follows: 100 mg of wet capsules were weighed, added to 10 mL of Sodium Citrate solution, 0.1 M (Merck, Spain), and shaken for 10 min to break the capsules and then sowed. Samples were diluted until appropriate concentrations were obtained before proceeding with the seeding in Petri dishes with MRS Agar (Merck, Spain) and incubated for 48 h at 37 °C. The microencapsulated bacteria were enumerated as CFU/mL.

The encapsulation yield (*EY*), which is a combined measurement of the efficacy of entrapment and the survival of viable cells during the microencapsulation procedure, was calculated according to following Equation (1) [22]:(1)$$EY=\left(\frac{N}{N_{0}}\right)\times 100$$
where *N* is the number of viable cells (log CFU g^−1^) released from the microcapsules and *N*_0_ is the number of viable cells (log CFU g^−1^) in the cell concentrate prior to microencapsulation.

### 2.4. FD Process

FD was carried out in a Telstar Lyo beta 25 equipment (Telstar, Madrid, Spain) with a fluidization temperature of 5.1 °C, a condenser temperature of −76.3 °C, and a heating temperature of 23.1 °C, with a full cycle of 48 h, beginning at a freezing temperature of −40 °C to 0.1 mbar (millibar) for 4 h. The entire process lasted 48 h.

### 2.5. FFD Strategies

FFD processes were carried out in a Telstar Lyo Quest equipment (Telstar, Spain). This process can be considered as a variant of FD, in which primary drying (elimination of approximately 90% of the water) is carried out through a succession of cycles of pressure changes (0.4–1000 millibars). Because of the ascending period of each cycle, heat accumulation is promoted (increase in the product temperature). This is followed by a rapid decrease in pressure with subsequent flash sublimation that returns the product to temperatures below 0° C.

In this work, three FFD strategies were applied, which differed basically in the temperature level reached in the product during primary drying. The first, identified as FFD-25, consisted of vaporization at temperatures between −25 °C and 0 °C; the second, FFD-15, consisted of temperatures between −15 °C and 0 °C; and the third one, FFD-3, consisted of temperatures between −3 °C and 0 °C. For the process, the heating plate temperature (20–55 °C) and pressure (0.4–1000 millibars) were regulated. Next, secondary drying was applied when the product temperature started to rise under the vacuum conditions, considered as the minimum achievable in the drying equipment used in the experiment.

A counting analysis of the probiotic living cells (viability test) was used as quality parameter to evaluate the suitability of the FFD process. The viability of the cells was analyzed before and after drying and expressed as survival rate (%) of the microorganisms after FFD strategies and can be calculated according to the method described in point 2.5. Cell viability of dried capsules was calculated as described in Section 2.3.

Water activity (*a_w_*) was determined in a hygrometer Novasina (Lachen, Switzerland) and was measured after each drying process.

Both FFD and FD bacteria were vacuum-packed and stored in the dark at 20 °C with 34% relative humidity for 28 days. During the storage time, cell viability measurements were also carried out every seven days.

### 2.6. Modeling of FFD

To evaluate the rate at which the water contained in the probiotic material has been eliminated during the different stages of drying for each FFD strategy, a mathematical model was developed based on the definition of the material and energy balances. These equilibria are applied to a microencapsulated probiotic material of spherical shape, which is arranged in the form of a bed of a certain thickness on a heating plate. Figure 1 shows the details of a differential element, *dz*, of layer, *L*, as well as its characteristic properties. The material to be dried is treated as a solid layer of porosity *ε*, whose voids are occupied by air.

#### 2.6.1. Mass Balance

Profiles of the mass loss of the encapsulated material in different phases of the process of drying are defined according to the following equation:(2)$$\varepsilon_{s\;}\rho_{s\;}\frac{\partial M_{z}}{\partial t}=\frac{\varepsilon \;m_{w}}{RT}D_{c}\frac{\partial^{2}P_{w}}{\partial z^{2}}+\frac{{\begin{array}{c} \left(1-\varepsilon \right)\;m \end{array}}_{w}}{RT}k_{c}\frac{\partial \left(P_{w}-P_{o}\right)}{\partial z}$$
where *P_o_* is the condenser pressure, and *P_w_* represent the vapor pressures of the material being dried. The difference between these two pressures is known as the driving gradient, which basically defines the drying rate. *P_w_* depends on the temperature and humidity, *M_z_*, at a given position, *z*, of the product.

Given the dependence of water activity on product moisture, the vapor pressure or driving gradient of the process will depend on the water activity and the saturation pressure according to Equation (3):(3)$$P_{w,z}=f\left(T_{z},M_{z}\right)=a_{w}P_{S}$$

#### 2.6.2. Energy Balance

The heat balance applied to a layer of microencapsulated product is expressed by Equation (4) in which the net heat flux (right-hand member) is divided into the energy required for the vaporization or sublimation of the water and heating or cooling of the product.
(4)$$\varepsilon_{s\;}\rho_{s\;}\frac{\partial M_{z}}{\partial t}\lambda_{s}+\rho_{L}C_{p}\frac{\partial T_{z}}{\partial t}=k_{t,L}\frac{\partial^{2}T}{\partial z^{2}}$$

Note that the drying rate ($\partial M_{z}/\partial t$) must simultaneously satisfy the material balance and the energy balance. The thermal conductivity of the material as well as the specific bed heat and the bed density are defined by the following equations:(5)$$\rho_{L}=\varepsilon_{w}\;\rho_{w}+\varepsilon_{s}\;\rho_{s}$$
(6)$$C_{p,L}≃\varepsilon_{w}C_{p,w}+\varepsilon_{s}\;C_{p,s}$$
(7)$$k_{t,L}=\frac{1}{\frac{\varepsilon_{w}\;}{k_{t,w}}+\frac{\varepsilon_{s}\;}{k_{t,s}}+\frac{\varepsilon \;}{k_{t,air}}}$$

In the above equations, the influence of the constituent parts of the material layer through the volume fraction of each of the three parts can be seen: air (ε), solid ($\varepsilon_{s}$), and water ($\varepsilon_{w}$). They are related according to Equation (8). Note that the bed porosity, *ε*, increases from its initial value, $\varepsilon_{L}$, because of the decrease in $\varepsilon_{w}$, due to the loss of water during drying, while $\varepsilon_{s}$ remains constant.
(8)$$\varepsilon +\varepsilon_{s}+\varepsilon_{w}=1$$

#### 2.6.3. Moisture Content and Drying Kinetics

Suppose *m_o_* is the mass of capsules charged on the heat plate of area, *A*, and a layer thickness, *L*; i.e., $m_{o}=A\;L\rho_{L,o}$. The solution of Equations (2) and (4) gives the moisture distribution *M_z_*, of capsules at different *z* positions. Thus, in a certain time, *t*, the integration of the moisture *M_z_* to the full thickness *L* of the bed distribution in the capsule leads to the corresponding average moisture *M_av_* through the following equation:(9)$$M_{av}\left(t\right)=\frac{\int_{0}^{L}M_{z}dz}{L}\;$$

The drying rate corresponding to an initial mass, *m_o_*, is defined by Equation (10), where $m_{s}\left(=A\;L\varepsilon_{s\;}\rho_{s\;}\right)$ is the mass of solid corresponding to a mass of loaded capsules.
(10)$$\frac{dm}{dt}=m_{s\;}\frac{dM_{av}}{dt}$$

The following assumptions were used to implement the model described above: mass and heat fluxes are unidirectional (moisture and temperature gradients are assumed only in direction *z*), there is no moisture gradient inside the capsules (the moisture on the capsule surface is the same as that inside), water only leaves as vapor, and from the capsule surface, particle shrinkage is negligible and water pressure $P_{w,z}$ is in equilibrium depending on the *T* and *M* of capsules at each position *z* [23,24,25].

### 2.7. Statistical Analysis

Statistical analyses were carried out in this work by applying a one-way analysis of variance (ANOVA) and Tukey’s test to the data with a significance level of *p* < 0.05 using software package Stat graphics Centurion version XVII.1.12 (StatPoint Technologies Inc., Warrenton, VA, USA). Data from triplicate experiments are presented as means with standard deviations.

## 3. Results and Discussion

### 3.1. Microencapsulation Process

Figure 2 shows photos of the LA microcapsules obtained in the microencapsulation process and the particle size deviation. The obtained mean diameter value of the microcapsules was 300 μm. The values obtained in this research are below the maximum capsule diameter size for oral administration (1 mm), thus achieving adequate palatability. The size of the capsule diameter depends on the microencapsulation technology, its efficacy, as well as the type of materials used and the methodology [26]. The use of sodium alginate and chitosan has been reported to be one of the most protective combinations for *Lactobacillus* stability because it favors its delivery to the colon at appropriate levels and aids in the maintenance of its survival during gastric and intestinal juice simulation [27].

### 3.2. Performing the FFD Experiments

The typical freeze-drying process consists of 3 steps: freezing, primary drying, and secondary drying. During these 3 steps, cells are exposed to various stresses, especially dehydration, compromising cell survival [28]. During FD the following process occurs: freezing at low temperatures below the melting point, drying by sublimation at a pressure below that corresponding to the triple point and desorption of the bound water [29]. Primary drying is the first drying stage of the freeze-drying process, in which the pressure is reduced, and sufficient heat is supplied to remove frozen water by sublimation, while secondary drying is the second drying stage, in which non-frozen water is removed by sublimation and the temperature is usually higher than in the primary drying stage.

The variation of the FFD strategies with respect to FD is that the drying of the probiotic material was done by intensified heating during primary drying, without melting the product, based on a cyclic change of pressure. Consequently, average minimum temperature levels of −25 °C, −15 °C and −3 °C are achieved for strategies FFD-25, FFD-15 and FFD-3, respectively. In all cases, the process was closed with a secondary drying in which there is a progressive convergence between the product temperature and the heating plate temperature, which takes place at 20 °C, from a pressure control. The total duration of each drying strategy was 900 min, which is 3.2 times less than the typical freeze-drying time of 2880 min, which is a dramatic advantage in terms of process efficiency.

#### 3.2.1. Freezing Temperatures Control

The results of the FFD-25 strategy are presented in Figure 3. In this case, the effect of applying the average minimum freezing temperature level of −25 °C to the product during primary drying for 160 min was evaluated. At the end of this first step of the process, 90% of the water was removed. In this context, secondary drying comprises two zones: one below 0 °C and the other above this temperature, starting at the moment when the product temperature begins to rise under the minimum vacuum power achievable using the drying equipment. The total duration of this step was 380 min. The third and last stage, which completed the secondary drying step, corresponded to a drying temperature ranging from 0 °C to 20 °C ± 1 °C with a duration of 360 min. This last stage, which completed the drying strategy, ended when the temperature of the product was equal to the temperature of the tray.

The results of the FFD-15 strategy are presented in Figure 4. The primary drying of this strategy was carried out with an average minimum freezing temperature of −15 °C with a duration of 150 min. The secondary drying began with a sustained increase in the product temperature from −7.7 °C to 0 °C when the first part of this step was finished, which lasted 50 min. Finally, the third phase, which completed the secondary drying step, corresponded to a product temperature ranging from 0 °C to 20 ± 1 °C and lasted 700 min when the product temperature was equal to the tray temperature.

In Figure 5, the results of the FFD-3 strategy are presented. In this process, primary drying started with the application of a vacuum and an average minimum product temperature of −3 °C. In this case, primary drying lasted only 80 min and was followed by secondary drying, which started with a sustained increase in the drying temperature from −3 °C to 15 °C, lasting 200 min for the first step. From this point on, we continued with the second part of the secondary drying process until the product temperature reached 20 ± 1 °C. The total time of the FFD-3 strategy was 900 min. This case is substantially different from the previous ones because the product temperature is quite close to 0° C at the beginning of the secondary drying step. Therefore, part of the combined water does not freeze at the beginning of this stage, so the end of the first part of the secondary drying step is observed with a pressure drop, when only a small part of the water, corresponding to 2%, remains to be vaporized.

The freezing stage is very important as it is the first stage of the process that drives not only the subsequent stages’ performance (i.e., primary and secondary drying), but also the quality of the product (i.e., physical stability, residual moisture, reconstitution time, among others). With respect to cell viability, some authors have pointed out that low cooling rates sharply reduce the viability of *Lactobacillus* [30]. The reduction in the freezing time in FFD strategies with respect to the FD process is also very important since the material may be partially frozen at this stage, with the freeze-concentrated fraction containing proteins in a liquid state. Protein molecules are susceptible to various stresses during this period, including, for example, the freezing concentration, changes in pH and ionic strength, interface, and local mechanics, among others. In this way, the risk of protein destabilization due to freezing stress can be reduced.

Likewise, FD, as a time- and energy-intensive process, requires a significant heat input to sustain water sublimation, which accounts for about 45% of the energy input in freeze-drying, whereas maintaining the vacuum environment takes an additional 26% of energy use. In FFD, the energy input is drastically lower, since the freezing temperatures are higher, and pressure is applied by reduced cycles in a short time. In addition, the increase in the shelf temperature leads to an increase in heat transfer rate to the product and a higher vapor sublimation rate. If the sublimed vapor cannot be removed fast enough by the flow to a lower-pressure region of the condenser, the vapor pressure in the chamber will rise [31].

#### 3.2.2. Pressure Control

Figure 6 describes the progress of the pressure applied during the three FFD strategies. Pressure regulates the vapor transfer rate, and it is possible to quickly modify the product temperature, especially in the primary drying step. During this period, a pressure variation is applied cyclically with a duration of approximately 10 to 20 min in the FFD-25 and FFD-15 experiments depending on the minimum temperature reached in the product with the pressure decrease. Thus, in the FFD-25 process, the cycles are of somewhat longer duration and in greater number than in the FFD-15 process. The FFD-3 case is a special situation given that the temperature of −3 °C is very close to the melting temperature, and it is not possible to perform cycles of pressure variation, so we decided to keep the pressure at the value necessary to maintain the product temperature with small variations between −3 and 0 °C.

Residual moisture at the end of drying or water activity is a decisive parameter in the quality of the final dehydrated product that can influence the final viability of the LA during storage. To this end, a study was carried out to analyze the change in water activity (*a_w_*) with the moisture content *M* of the product. Water activity measurements were made at different times during the drying process. Triplicate samples were taken on the initial sample at three moments of the primary drying step and towards the end of the first part of the secondary drying step. The results can be seen in Figure 7, which shows the mean value and the observed deviation of *a_w_* and *M* in the measurements taken, indicating the water activity of the product, *a_w_*, as a function of a decreasing *M* value or, in other words, with an increasing drying time. It can be seen that the results correspond to primary drying, in which *a_w_* decreases smoothly with the aqueous content with a significant decrease in *M* of about 90% of the initial water content. In secondary drying, the drop in *a_w_* is more intense, distinguishing a first phase (“knee” in Figure 7) of moderate decrease, followed by a strong decrease until the final drying step.

It is important in the drying process to know the vapor pressure of the product, *P_w_*, in relation to the pressure of the drying chamber, *P*. Thus, *P_w_* was estimated at critical moments: at the end of primary drying (end P.D.), the end of the first phase of secondary drying (end S.D1), and the end of drying (final); see Table 1. The difference between *P_w_* and *P* gives us a quick idea of the intensity of the drying rate, which is greater as greater the difference. During the FFD experiments studied, there are situations of greater or lesser duration in which the *P_w_*, due to the temperature of the product and its moisture content, can become higher than the total pressure in the chamber. This situation can cause a sudden (flash) sublimation or vaporization of the moisture, depending on the product temperature, which is much more intense than when the difference is in favor of *P*. These situations in which *P_w_* exceeds *P* are characteristic of the FFD process, and they can accelerate the drying process and even cause a certain expansion in the product like other drying modalities such as puffing drying [32]. Thus, FFD modalities become interesting alternatives to elaborate dehydrated snack-type special foods, with the probiotic material content being an added value [33]. The products can present a certain degree of swelling depending on the drying conditions and constitution of the material. Consequently, products with more porous structures present less difficulty for the elimination of moisture during secondary drying, facilitating the achievement of the ideal amount of moisture for the stabilization of the dehydrated product during storage.

Table 1 shows the profile of the vapor pressure, *P_w_*, through the values at the most significant intermediate points, such as the end of primary drying and the end of secondary drying (first part) and the end of the drying process. The FFD-3 case presents a special situation in which the difference of *P_w_* over *P* is small but remains uninterrupted throughout the primary drying and with *P_w_* values that are clearly higher than those in the other two experiments. Therefore, the drying rate in the primary drying step, although maintained, is less intense, and that seems to be the cause of the water activity being higher than that of the other two cases at the end of FFD-3 drying.

### 3.3. Drying Kinetis Analysis

This section analyzes the importance that the drying process plays in the stabilization of an encapsulated product through the drying rate at which the moisture is removed from the microencapsulated probiotic material. As indicated in the previous sections, basically, three variants of freeze-drying known as FFD are studied, in which the operational changes, with respect to conventional freeze-drying (FD), can be transcendental on the quality of the dehydrated product and along the storage. If we compare the operating conditions applied in FFD with those of FD, pressure stands out as the most important operational variable, while in FD, the pressure remains constant. The up–down changes in pressure that occur in FFD during primary drying have a special impact on the drying rate.

In this work, an estimation of the drying rate in the three FFD strategies studied was carried out by applying the mathematical model described in Section 2.2 above. Primary drying involves the elimination of a high percentage of the aqueous content, which can be up to 90%, depending on the type of product, and is basically made up of free water [34].

The FFD experiments were carried out on a quantity of microencapsulated material of about 250 g that was placed on a heating surface of 295 cm^2^. The thermophysical properties of the product, as well as the heat and mass transport parameters of the mathematical model described in Section 2.2, were estimated based on a previous work [35]. For the simulation of the FFD experiments, a calculation program was used with a proprietary software developed in Python [20]. The program was fed with the data obtained in the FFD experiments: the profiles of product temperature, heating plate temperature, pressure, and process duration. Consequently, the simulation program provides, as a result, the drying rate and mass loss along each FFD experiment.

Figure 8 shows how the pressure drops observed in Figure 6 are associated with high drying rates. In addition, the intensity of drying also appears to be related, although not in the same way, to the material temperature that needs to be maintained during drying. Consequently, FFD-15 provides the highest point drying rate (19.61 g/min), followed by FFD-25 and FFD-3, as can be seen in Figure 8a,b,d. By comparing the maximum values of the drying rate, we can deduce that, after the maximum, the following high-intensity instants are characterized by a continuous decrease in intensity because of water loss. These instants of high drying intensity (flash drying), together with the temperature, seem to be essential in the effect on the porosity and quality of the final dried product. Thus, FFD-25, without having the maximum drying rate, is the one that, as will be seen later, presents the best results on LA viability, not only at the end of drying but also during storage. Figure 8a,c shows the correspondence of the moment (45 min) of maximum speed with a strong drop in mass of 50 g in less than 10 min. The uniform distribution of the remaining “saw-teeth” moments would provide a porous structure that facilitates complete moisture removal in secondary drying.

### 3.4. Effect of Drying Strategies on Bacterial Viability

In Figure 9, the viability cells after and before the FFD strategies are shown.

According to these results, all freeze-drying variants showed a decrease in cell viability after application. Although the cell viability presented by FD and FFD-25 are comparable (Log CFU/g 7.85 ± 0.01 and 7.94 ± 0.01) (no significant differences, *p* < 0.05), it is noteworthy that the freezing temperature of −25 °C reported the highest cell viability of 89.92%, which is 2.74% higher than that obtained in the freeze-drying process, which was 87.52%.

The cell viability levels found in this research agree with those reported by other authors, in relation to the survival of freeze-dried *Lactobacillus*, who report a range of 86.82–93.98% being the highest values obtained when skim milk powder was used as a cryoprotective agent [36], which was also used in this research. Among the benefits associated with skim milk as an effective cryoprotectant, it has been noted that it stabilizes cell membrane constituents and prevents cell damage [37].

Although the resistance of freeze-dried probiotic microorganisms to process temperatures is related to the genus and strain of the microorganism, it is important to note that optimal temperatures above −25 °C have been reported during the optimization of sublimation conditions. Thus, for freeze-dried *Bifidobacterium longum* BB68S, the temperature at which the highest survival rate was recorded was −10 °C (65.4 ± 3.8%) compared to −20 °C (42.3 ± 3.8%) and 0 °C (56.4 ± 2.2%) [38].

In the present work, the freezing temperature of −3 °C recorded the lowest cell viability of log CFU/g of 7.34 ± 0.03, which represented 83.79%. Other authors selected −30 °C as the most favorable temperature for freezing *Lactobacillus acidophilus* KBL409 encapsulated in alginate–chitosan, reporting that the cells were minimally affected by the solution and mechanical effects as well as intracellular ice formation [39].

The concepts of FFD applied in this work is based on the speed of drying, and freezing has been implemented rapidly. This rapid implementation is considered a fundamental parameter for preventing crystal formation in probiotic materials. The freezing step was carried out in a short time thanks to frequent pressure changes that cause significant alterations in the drying speed compared to the conventional method. However, it must be noted that the optimal freezing rate could varies from genus to genus [40].

It has also been reported that optimizing the pre-freezing temperature can significantly improve the survival rate of a probiotic strain. A low pre-freezing rate causes intracellular water to flow into the extracellular medium and causes extracellular crystals to form. In contrast, a high pre-freezing rate could cause an insufficient flow of intracellular water into the extracellular medium, and the formation of intracellular crystals that are too large could severely damage the cell walls of the strains. It has been reported that more severe cell membrane damage occurred at −196 °C than at −20 °C applied to *L. plantarum* AR113 as pre-freezing temperatures in the freeze-drying process, which is also influenced by the cryoprotectant used [41].

### 3.5. Effect of Drying Strategies on Water Activity

The results of dried LA’s *a_w_* are presented in Figure 10.

FFD-25 presented the lowest value of water activity (*a_w_*) of 0.0522 ± 0.002 (significant differences, *p* < 0.05), which is 55% lower than the value presented in the product treated with the FD control, which was 0.0941 ± 0.003, showing that the freezing temperature of −25 °C was the most effective in removing the bound water contained in the microencapsulated LA. Also, FFD-15 and FFD-3 presented lower a_w_ values than the FD control (no significant difference, *p* > 0.05).

It has been reported that storage at very low temperatures is better than at higher temperatures to obtain freeze-dried bacteria with high stability. In addition, it has been observed that the viability of freeze-dried lactobacilli depends on the *a_w_*. Thus, by evaluating freeze-dried *L. acidophilus* with different water activities (0.11, 0.22, 0.32, and 0.43), it was recorded that, in general, increasing water activities decreased the survival of the bacteria during 12-week storage at 30 °C [42].

Also, it has been suggested that a_w_ values above ~0.25 are likely to increase the mortality rate of probiotic bacteria during storage due to increased bacterial metabolism. The water activity values found in this investigation by both FFD and FD are significantly below this critical limit. Likewise, some authors have reported values of 0.14 ± 0.01 for freeze-dried *Lactobacillus plantarum* processed at −55 °C for 2 days [26], suggesting that the FFD process has significant advantages over the traditional freeze-drying process for probiotic Lactobacillus. *a_w_* values lower than those found in this investigation have also been reported for *Bifidobacterium longum* BB68S (0.012–0.020) [38]. The water content is important for probiotic product storage [43,44], and it is recommended that *a_w_* should be in the optimal range. Likewise, water can interact with functional groups and block reaction sites, thus avoiding undesirable degradation reactions in cells [45].

### 3.6. Analysis of LA Viability during Storage

Figure 11 shows the survival rate of LA during the storage time for the three freezing temperatures (−25 °C, −15 °C, and −3 °C) and for the control, FD.

In all drying treatments, the decrease in cell viability was sustainable during storage. The most pronounced decrease in viability was observed at 7 days, followed by 14 days (significant differences (*p* < 0.05), with respect to the other storage times (21 and 28 days), showing more moderate and similar decrease curves during 21 and 28 storage days. In this work, the greatest impact on the decrease in cell viability occurred during the first 14 days of storage, at a temperature of 20 °C, well above the refrigeration temperatures (5 °C) recommended for the storage of freeze-dried lactobacilli to preserve cell viability, significantly increasing the cost of commercialization.

However, it is noteworthy that, at the end of storage, the highest viability was obtained in FFD-25, with a level of 4.98 log CFU/g (64.72% cell survival), which is slightly above that obtained by the control FD treatment of 4.82 log CFU/g, which represented only 61.03% cell survival (no significant difference, *p* > 0.05).

These results are comparable with those obtained by Tang et al. [46], who observed reductions in cell viability from 13.01 log CFU to 8.71 ± 0.22 (66.95% cell survival) and to 7.88 ± 0.27 (60.57% cell survival) in freeze-dried *Lactobacillus acidophilus* FTDC 3081 treated with skim milk as a cryoprotectant stored for 30 days at temperatures of 25 °C and 40 °C, respectively. Among the factors that favor the cell viability of lactobacilli during storage are the use of cryoprotectants and antioxidants, since they protect it from cell stress caused by the temperature process, as well as from oxygen and environmental humidity, among other factors [47,48]. In this work, correspondence was observed between the dehydrated lactobacilli that obtained lower water activity with those that presented higher levels of cell viability during the storage studied, which is in agreement with the results reported in different studies [49,50].

## 4. Conclusions

Flash freeze-drying (FFD) presented in this work is a variant of strict freeze-drying (FD) that increases the freezing rate during primary drying by abrupt pressure changes consisting of rapid pressure rises and falls. FFD lasted for a total of 900 min, which is 68.75% less than the usual freeze-drying time of 2880 min, with a consequent drastic decrease in the exposure time of the freeze-dried bacteria to temperature extremes that could affect their cellular integrity.

Among the three freezing temperatures studied during primary drying (−25 °C, −15 °C, and −3 °C), −25 °C showed, for *Lactobacillus acidophilus* LA5 (LA), a cell viability of 89.94%, which is 2.74% higher than that obtained in FD, and a water activity of 0.0522, which is significantly lower than that recorded by FD. Likewise, at the end of the storage time (28 days/20 °C and 34% relative humidity), it showed the highest cell survival rate of 62.72%, which is 2.78% higher than that of FD.

Based on the experimental data, a mathematical model was developed to obtain the optimal operating parameters like drying rate and mass loss along each FFD experiment. FFD-25 showed a maximum 9.85 g/min drying rate, that leaded to the best results of LA viability in the final dried material.

The results presented in this work reveal that FFD could be an effective drying technique to preserve the viability of microencapsulated probiotic microorganisms, with the advantage of a drastic reduction in operation time and costs, compared to traditional FD.

Further studies should be conducted with other genes and strains to define the most favorable processing temperature for each one.

## Figures and Tables

**Figure 1 microorganisms-12-00506-f001:**
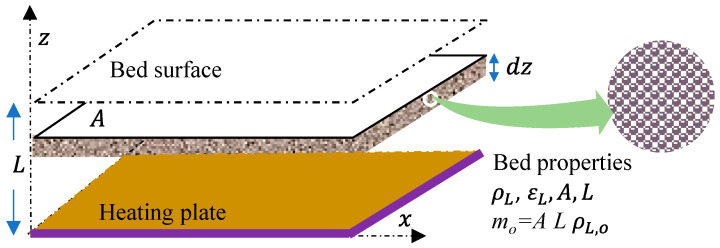
Schematic of a bed of microencapsulated particles showing geometrical variables and material properties.

**Figure 2 microorganisms-12-00506-f002:**
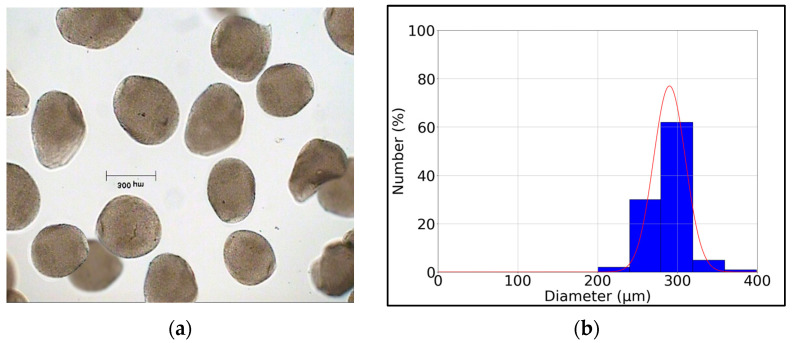
(**a**) Encapsulated *Lactobacillus acidophilus* LA. (**b**) Particle size distribution.

**Figure 3 microorganisms-12-00506-f003:**
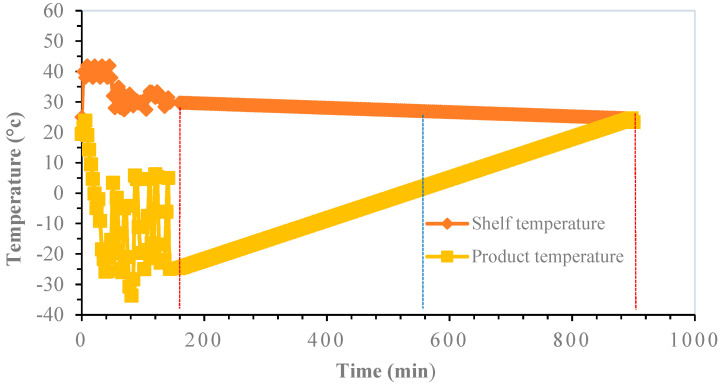
FFD-25 strategy. Primary drying was carried out from 0 to 160 min (red vertical line). Secondary drying (1st stage) was carried out from 160 to 540 min (blue line). Secondary drying (2nd stage) from 540 to 900 min (2nd red line).

**Figure 4 microorganisms-12-00506-f004:**
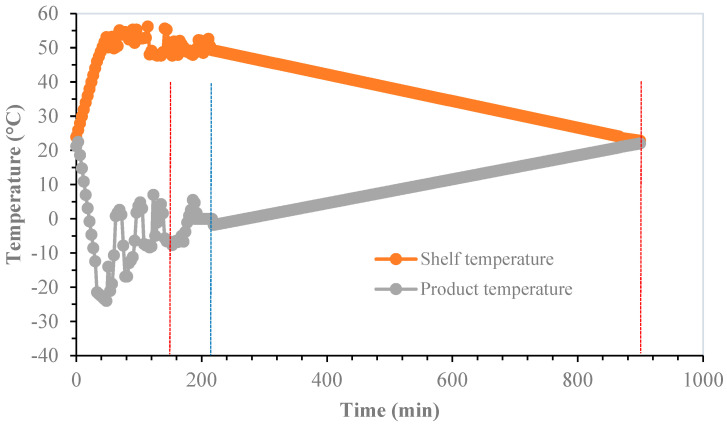
FFD-15 strategy. Primary drying from 0 to 150 min (red vertical line). Secondary drying (1st stage) from 150 to 215 min (blue line). Secondary drying (2nd stage) from 215 to 900 min (2nd red line).

**Figure 5 microorganisms-12-00506-f005:**
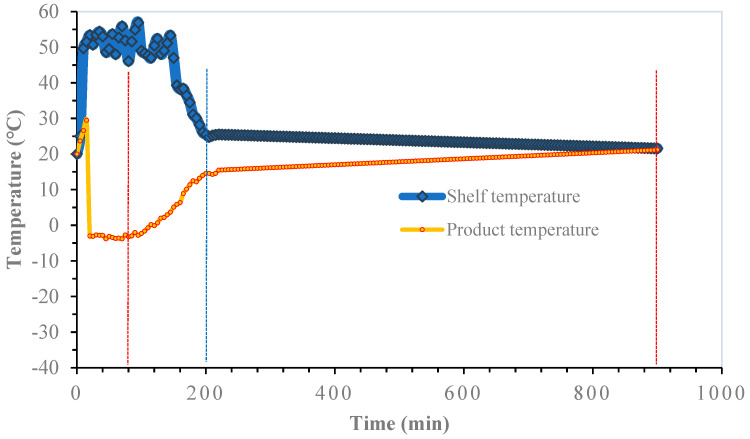
FFD-3 strategy. Primary drying from 0 to 80 min (red vertical line). Secondary drying (1st stage) from 80 to 200 min (blue line). Secondary drying (2nd stage) from 200 to 900 min (2nd red line).

**Figure 6 microorganisms-12-00506-f006:**
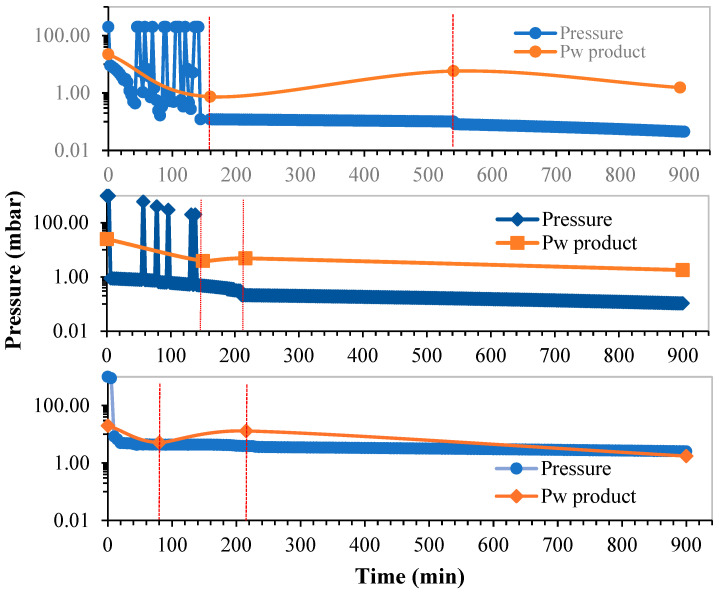
Pressure variation in the FFD experiments performed: FFD-25 (**top**), FFD-15 (**medium**), and FFD-3 (**bottom**). The orange line shows the vapor pressure profile of the product, *P_w_*, through the intermediate critical points of drying indicated with vertical lines.

**Figure 7 microorganisms-12-00506-f007:**
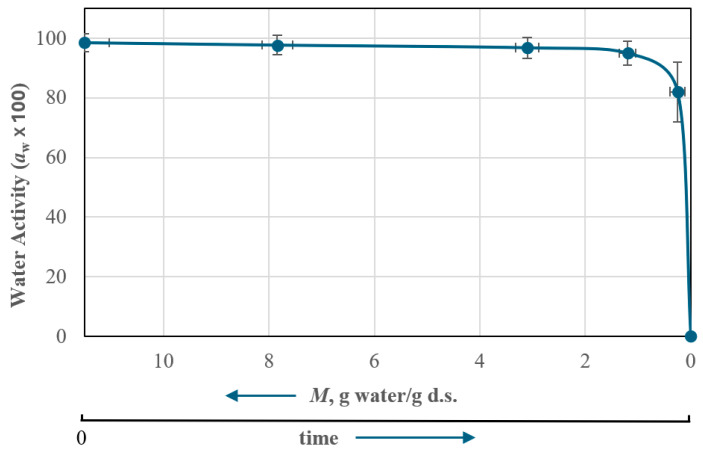
Variation in water activity, *a_w_*, as a function of moisture content of microencapsulated probiotic material.

**Figure 8 microorganisms-12-00506-f008:**
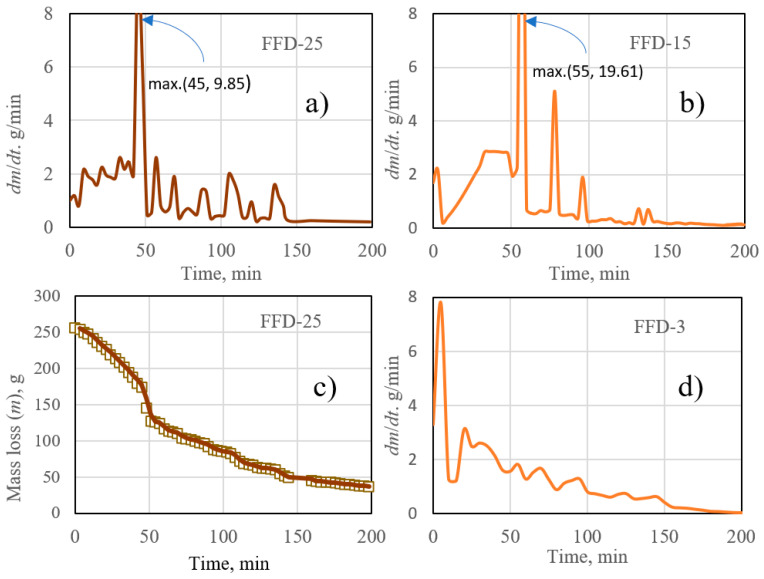
Drying rate profiles obtained in the simulation of FFD experiments: (**a**) FFD-25 case, (**b**) FFD-15 case, (**d**) FFD-3 case, and (**c**) mass loss of FFD-25.

**Figure 9 microorganisms-12-00506-f009:**
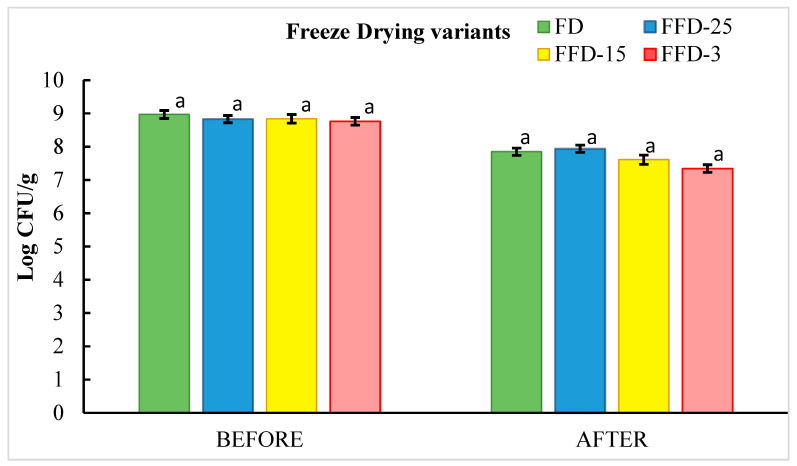
LA cell viability before and after different drying variants (FFD and FD) and times. (Same letters means non-significant differences, *p* > 0.05).

**Figure 10 microorganisms-12-00506-f010:**
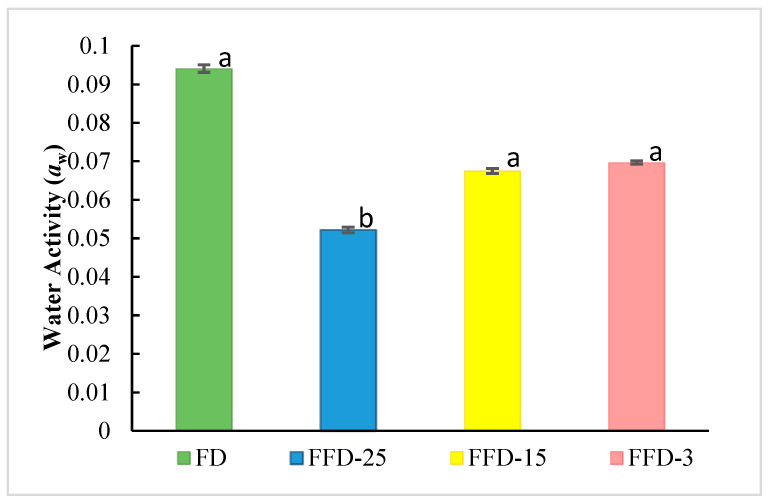
Water activity of encapsulated LA at the end of different drying variants. (Different letters mean significant differences, *p* < 0.05).

**Figure 11 microorganisms-12-00506-f011:**
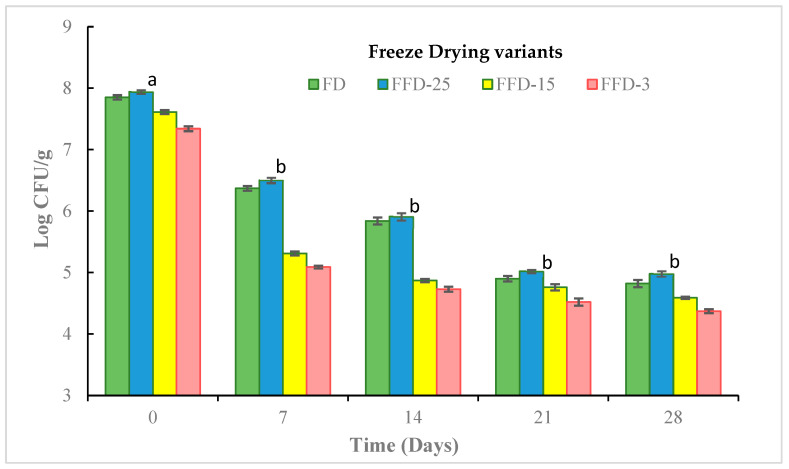
LA cell viability during storage time. (Different letters mean significant differences, *p* < 0.05).

**Table 1 microorganisms-12-00506-t001:** Values of humidity *M* and product temperature, *T* p., for the calculation of the vapor of product *P_w_* at different critical points of the FFD experiments (P.D.: primary drying; S.D1.: secondary drying 1st part; Final: end of drying process).

Critical Times	FFD Experiment	*M*	*a_w_*	*T* p.	*P* _s_	*P_w_* = *P_s_ *∗ *a_w_*
End P. D.	FFD-25	1.15	0.949	−24.5	0.79	0.75
	FFD-15	1.15	0.949	−7.7	4.13	3.92
	FFD-3	1.15	0.949	−2.6	5.44	5.17
End S. D1.	FFD-25	0.23	0.810	0.7	7.15	5.79
	FFD-15	0.23	0.810	−0.5	5.98	4.85
	FFD-3	0.23	0.810	14.6	16.20	13.12
Final	FFD-25	0.11	0.052	24.4	29.50	1.54
	FFD-15	0.10	0.068	22.0	26.72	1.80
	FFD-3	0.19	0.074	21.0	25.05	1.75

## Data Availability

Data are contained within the article.

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
