# Peer review of "Examining the Effect of Freezing Temperatures on the Survival Rate of Micro-Encapsulated Probiotic Lactobacillus acidophilus LA5 Using the Flash Freeze-Drying (FFD) Strategy"

_microorganisms, 2024, doi:10.3390/microorganisms12030506_

Round 1

Reviewer 1 Report

Comments and Suggestions for Authors

The conducted studies are very interesting and innovative. However, the discussion of the authors' own results lacks comparison with the results of other authors. In addition, the authors wrote that statistical calculations were carried out, however, there is no table with the results, only references to ANOVA are included in the text.

Reviewer 2 Report

Comments and Suggestions for Authors

1. Improvements are needed for Figure 6 and Figure 1b to enhance readability.

2. The introduction and analysis sections should include more discussions on other relevant studies.

3. How were the three temperatures (-25°C, -15°C, and -3°C) determined in this study?

4. The format of the figures should be consistent, such as the font and size in Figures 3-5 and the vertical axis font in Figures 8a and 8b.

5. The significance test results in Figures 9 and 10 should be labeled in the figures.

6. It seems that in Line 71, "Lactobacillus acidophilus LA5 (LA5) L was supplied " should be changed to "Lactobacillus acidophilus LA5 (LA) was supplied".

7. Please provide specific data (mean and standard deviation) of Figure 11 for review.

8. The title of Table 1 needs to be more specific, indicating F.D. and S. D1.

9. The relevant content in the MS needs to be consistent, such as Line 75 and 76, "minutes" and "min"; Line 139 "aw"; Line 77 ""; Line 272 and 273 1st and 2nd

; and Line 376 "cm2".

10. There are some unusual spaces in the MS that need to be confirmed, such as Line 67, 81, 141, 152, and 193.

11. In Line 72, why is it necessary to specify "Celsius" when "" has already been used earlier?

12. In Line 484, does the MS have any Supplementary Materials?

13. The formatting of the references needs to be adjusted for consistency.
